# Designing Effective Multi-Target Drugs and Identifying Biomarkers in Recurrent Pregnancy Loss (RPL) Using In Vivo, In Vitro, and In Silico Approaches

**DOI:** 10.3390/biomedicines11030879

**Published:** 2023-03-13

**Authors:** Andrés Alexis Ramírez-Coronel, Amirabbas Rostami, Laith A. Younus, José Luis Arias Gonzáles, Methaq Hadi Lafta, Ali H. Amin, Mohammed Abdulkadhim Saadoon, Hayder Mahmood Salman, Abolfazl Bahrami, Rossa Feilei, Reza Akhavan-Sigari

**Affiliations:** 1Epidemiology and Biostatistics Group, Research Group in Educational Statistics, National University of Education (UNAE), Azogues 030102, Ecuador; 2Azogues Campus Nursing Career, Health and Behavior Research Group (HBR), Psychometry and Ethology Laboratory, Catholic University of Cuenca, Cuenca 010109, Ecuador; 3Psychology Group, University of Palermo, Buenos Aires 60301, Argentina; 4Epidemiology and Biostatistics Research Group, CES University, Medellín 050001, Colombia; 5Department of Internal Medicine, Faculty of General Medicine, Yerevan State Medical University, Yerevan 375010, Armenia; 6Department of Clinical Laboratory Sciences, Faculty of Pharmacy, Jabir Ibn Hayyan Medical University, Al Najaf Al Ashraf 54001, Iraq; 7Pontificia Universidad Católica Del Peru, Lima 15001, Peru; 8Iraqi Ministry of Education, Baghdad 10011, Iraq; 9Deanship of Scientific Research, Umm Al-Qura University, Makkah 21955, Saudi Arabia; 10Zoology Department, Faculty of Science, Mansoura University, Mansoura 35516, Egypt; 11Directorate General of Education Karkh 1, Ministry of Education, Baghdad 10011, Iraq; 12Department of Computer Science, Al-Turath University College Al Mansour, Baghdad 10011, Iraq; 13Biomedical Center for Systems Biology Science Munich, Ludwig-Maximilians-University, 80333 Munich, Germany; 14Department of Cell Biology, Tuebingen University, 72072 Tuebingen, Germany; 15Department of Neurosurgery, University Medical Center, 72072 Tuebingen, Germany; 16Department of Health Care Management and Clinical Research, Collegium Humanum Warsaw, 04080 Warszawa, Poland

**Keywords:** recurrent pregnancy loss, hub genes, hub high traffic gene, RPL, biomarkers, polo-like kinase 1, NF-κB signaling pathway

## Abstract

Recurrent pregnancy loss (RPL) occurs in approximately 5% of women. Despite an abundance of evidence, the molecular mechanism of RPL’s pathology remains unclear. Here, we report the protective role of polo-like kinase 1 (PLK1) during RPL. We aimed to construct an RPL network utilizing GEO datasets and identified hub high-traffic genes. We also investigated whether the expressions of PLK1 were altered in the chorionic villi collected from women with RPL compared to those from healthy early pregnant women. Gene expression differences were evaluated using both pathway and gene ontology (GO) analyses. The identified genes were validated using in vivo and in vitro models. Mice with PLK1-overexpression and PLK1-knockdown in vitro models were produced by transfecting certain plasmids and si-RNA, respectively. The apoptosis in the chorionic villi, mitochondrial function, and NF-κB signaling activity was evaluated. To suppress the activation of PLK1, the PLK1 inhibitor BI2536 was administered. The HTR-8/SVneo and JEG-3 cell lines were chosen to establish an RPL model in vitro. The NF-κB signaling, Foxo signaling, PI3K/AKT, and endometrial cancer signaling pathways were identified via the RPL regulatory network. The following genes were identified: *PLK1* as hub high-traffic gene and *MMP2*, *MMP9*, *BAX*, *MFN1*, *MFN2*, *FOXO1*, *OPA1*, *COX15*, *BCL2*, *DRP1*, *FIS1*, *TRAF2*, and *TOP2A*. Clinical samples were examined, and the results demonstrated that RPL patients had tissues with decreased PLK1 expression in comparison to women with normal pregnancies (*p* < 0.01). In vitro, PLK1 knockdown induced the NF-κB signaling pathway and apoptosis activation while decreasing cell invasion, migration, and proliferation (*p* < 0.05). Furthermore, the in vivo model proved that cell mitochondrial function and chorionic villi development are both hampered by PLK1 suppression. Our findings revealed that the PLK1/TRAF2/NF-κB axis plays a crucial role in RPL-induced chorionic villi dysfunction by regulating mitochondrial dynamics and apoptosis and might be a potential therapeutic target in the clinic.

## 1. Introduction

Recurrent pregnancy loss (RPL) is the term used to describe spontaneous abortions that occur three times or more in a row before 20 weeks of pregnancy and includes embryonic or fetal loss; it is a frequently occurring human infertility-related disease that affects 1–5% of parturients [1]. Various factors have been proven to cause the occurrence and development of RPL, including chromosomal abnormalities, genital tract anomalies, immunological diseases, endocrine diseases, antiphospholipid syndrome, thrombophilic disorders, and pathogen infections [2]. Approximately 40–50% of cases remain unexplained, and the molecular mechanisms have not been fully explored. These cases are defined as unexplained recurrent pregnancy loss [3]. Although the diagnosis of RPL is relatively clear, the lack of standardized definitions, the uncertainties of its pathogenesis, and the variable clinical manifestations still hamper progress in the treatment and prevention of RPL [4]. The different occurrences that construct the complex gestation process include parturition, placentation, decidualization, and implantation [5]. As well, molecular and physiological processes must support a relationship between a receptive uterus and the implantation of the embryo. The hierarchical process of embryo implantation necessitates fundamental techniques including apposition, adhesion, attachment, and penetration, in which villous trophoblasts and high-quality embryos play a crucial role. During embryo implantation, extravillous trophoblasts (EVTs) of the decidua basalis originating from trophoblastic cell columns of anchoring villi invade the maternal uterine decidua, up to the inner third of the myometrium. This process, i.e., invasion of EVTs, facilitates the attachment of the placenta to the uterus (interstitial invasion) to provide nutrients to the embryo from the placenta [6]. Consequently, trophoblasts are essential for a healthy pregnancy because they are the placenta’s precursor cells. Recent research has shown that aberrant spiral artery remodeling and shallow placentation may both contribute to dysfunctional trophoblast functions that are ultimately linked to poor pregnancy outcomes such as stillbirth and abortion.

Moreover, recurrent pregnancy loss involves a cascade of physiological reactions as well as the activation of numerous signaling pathways, and the NF-κB (nuclear factor-κB) pathway is a crucial pathway. Although, the NF-κB pathway’s function in different biological processes is debatable; on the one hand, it can promote host defense and other protective cellular responses, while on the other, its activation contributes to inflammatory harm in tissues [7]. It is not unexpected that the NF-κB pathway impacts mitochondrial function and architecture as the NF-κB pathway serves as a transcription factor of the oxidative stress reaction [8]. However, additional research should be carried out on the NF-κB precise function in the fusion and fission of mitochondria in the epithelium during pregnancy.

The well-known mitotic regulator polo-like kinase 1 (PLK1) participates in G2/M transition, DNA integrity maintenance, and the overexpression maturation of various organelles [9,10,11]. It has been shown that PLK1 inhibits NF-κB nuclear activation, but the related mechanism is unexplored. One of the upstream regulators of NF-κB signaling is the TRAF (Tumor necrosis factor receptor-associated factor) family proteins such as TRAF2. A previous investigation showed that PLK1 and TRAF2 interact in HEK293 cells [12]. However, the biological role of PLK1 in trophoblasts and consequently in the occurrence and progression of RPL remains unclear.

Furthermore, omics studies generate a large amount of information concerning the biomarkers, molecular mechanisms, and biological pathways involved in complex diseases [13]. More recently, due to advances in the development and optimization of high-throughput techniques, numerous studies have applied omics approaches to the study of RPL [14,15,16,17]. Therefore, in the present study, we aimed to construct an RPL regulatory network and investigate whether *PLK1* as a hub high-traffic gene or protein induces a change in trophoblast functions in terms of proliferation, migration, invasion, and apoptosis and further contributes to RPL via the TRAF2/NF-κB axis. Additionally, we investigated whether PLK1 has an impact on early embryo growth and quality, affecting implantation and ultimately leading to RPL, using mouse pre-implantation embryos.

## 2. Materials and Methods

### 2.1. Data Mining and Identification of Genes Expression

RPL-related mRNA data of patient tissues and normal tissues were integrated from RNA from RNA-seq data expression datasets. In addition, the RNA-seq data were separately collected from the Gene Expression Omnibus (GEO) database (https://www.ncbi.nlm.nih.gov/geo/ (accessed on 14 Jul 2022)), and the three gene expression profiles (GSE180485 [13], GSE161969 [18], and GSE65102 [19]) were selected with the GPL11154 platform.

Then, the raw sequences quality was assessed utilizing FastQC (v. 0.11.11); then, these reads were trimmed employing Trimmomatic (v.0.39.2) to exclude the polymerase chain reaction (PCR) primers, low-quality reads, and adapters [20]; HISAT2 (v. 2.2.2) was utilized to map the trimmed reads to the reference genome of *Homo sapiens* [21]. Utilizing the DESeq2 program (v. 2.11.41) [22], variations in transcript expression were then discovered. The threshold of statistical difference for mRNA/gene expression was set using the parameters of logFC (|log2(FC)| > 1.5) and false discovery rate (FDR) < 0.05. A total of 143 samples from RPL groups and healthy controls were included in these datasets.

### 2.2. Recurrent Pregnancy Loss Regulatory Network Construction and Module Finding

Construction of the recurrent pregnancy loss regulatory network was performed utilizing different databases. This network was combined with BIND, PPI, and BioGRID databases [23]. Interaction data were also detected by seeking in interaction databases including the STRING (https://string-db.org/ (accessed on 25 July 2022)) and GeneMANIA (https://genemania.org/ (accessed on 25 July 2022)) databases. The PPI network was shown using the Cytoscape program (v. 3.7.2), and the hub genes were screened using Cytoscape’s “degree” value in the “cytohubba” plugin [24]. A module-screening technique that was contrasted was molecular complex detection (MCODE) [25]. The network entropies were determined to counteract the selective speculation [26].

### 2.3. Functional Enrichment Analyses

Applying the “ClueGO” package in R with a cut-off condition of the adjusted *p*-value less than 0.05, Kyoto Encyclopedia of Genes and Genomes (KEGG) pathway and GO (gene ontology) analyses were conducted [27]. Additionally, the Enrichr database was operated to conduct Wiki pathways of common host variables. The The Database for Annotation, Visualization and Integrated Discovery (DAVID) [28], g: Profiler (https://biit.cs.ut.ee/gprofiler/ (accessed on 10 August 2022)) [29], and GeneCards (www.genecards.org/ (accessed on 10 August 2022)) databases were utilized to investigate the pathways.

### 2.4. Samples Collection

Samples were collected from consenting participants from the Gynecology and Obstetrics Department of the King Faisal Hospital. The present study was reviewed and preapproved by the Ethics Committee of King Faisal Medical University (Approval number: PU-2021–11513, 29 June 2021) and all participants provided written informed consent. The control group included villi of the placenta taken from 26 healthy women between the ages of 19 and 40 who had already had one or more children but who choose to end their pregnancies on non-pathological bases. None of these ladies had ever previously experienced an abortion, a pre-term birth, or a stillbirth. The RPL group, on the other hand, consisted of fifteen patients between the ages of 19 and 40 who had gone through at least two consecutive first-trimester losses with an unknown cause between 2021 and 2022. This study excluded participants who had recognized risk factors for RPL, such as uterine deformity, hormone issues, significant chromosomal abnormalities, and infections. Appendix A provides a complete list of the study participants’ characteristics. In the current investigation, samples of villous were acquired using a suction technique that was carried out following routine strategies. Following dilatation and curettage, fresh villi tissues that were free of bleeding and calcification zones were collected within 10 min, stored on ice, and brought right away to the lab. Each sample was divided into smaller pieces, each of which was then dehydrated operating an ethanol solutions gradient before being ingrained in paraffin. The samples were immediately transferred to a nitrogen tank and kept at −196 °C.

### 2.5. RNA-Seq Validation

To obtain total RNA, a GenElute RNA Extraction Kit (Sigma, St. Louis, MO, USA) was used. Evo M-MLV Mix Kits were used to create single-stranded complementary DNA from whole RNA (Qiagen; Hilden; Germany). Utilizing SYBR green premix pro-Taq qPCR Kits, real-time PCR was carried out (Qiagen; Hilden; Germany). The following step involved doing quantitative reverse transcription PCR for five genes, including *TRAF2*, *PLK1*, *OPA1*, *FIS1*, *MFN1*, and *DRP1*. The PCR cycling conditions employed were as follows: 30 s at 95 degrees, as well as 40 cycles of 5 s each at 95 and 60 degrees. Finally, the relative expression of gene levels was normalized to the expression of *GAPDH*. The forward and backward sequence of primers and related genes ID are presented in Appendix A.

### 2.6. Immunohistochemistry

Villi tissue was embedded in paraffin, fixed in paraformaldehyde 4%, and then cut into 6 m pieces. Next, for antigen retrieval, rehydration, and deparaffinization, goat serum was used to prevent nonspecific sections. Antigen retrieval was then conducted following a pre-treatment in 12 mM sodium salts of citric that was microwaved for 15 min. Following that, 2% H_2_O_2_ was utilized to inhibit endogenous peroxidase. The slides were then exposed to an anti-PLK1 (1:500, Cambridge, UK) antibody for overnight incubation at 4 °C, followed by an appropriate biotin-conjugated secondary antibody. An Olympus optical microscope was used to image the slides’ complete field of view (Olympus, Tokyo, Japan).

### 2.7. Western Blotting

The villi tissue samples were isolated and a protein extraction kit (Beyotime Biotechnology, Shanghai, China) was used to extract whole proteins. Extracted proteins were rinsed with phosphate-buffered saline (PBS) and lysed with phosphatase/protease inhibitor cocktails. The protein concentrations were quantified by a BCA protein assay kit (Beyotime Biotechnology, Shanghai, China). The whole proteins were boiled and isolated in 12% SDS-PAGE and disseminated to polyvinylidene difluoride membranes and incubated at 4 °C utilizing antibodies PLK1 (1:1000; Cambridge, UK), OPA1 (1:500; Cambridge, UK), DRP1 (1:500; Cambridge, UK), MFN1 (1:500; Cambridge, UK), FIS1 (1:500; Cambridge, UK), TRAF2 (1:500; Cambridge, UK), NF-κB (1:500; Cambridge, UK), Foxo1 (1:500, Cambridge, UK), CDK6 (1:1000, Cambridge, UK), CycilneD1 (1:500, Cambridge, UK), and actin (1:2000; Cambridge, UK). Finally, the signals were detected by a digital imaging set, and ImageJ v. 2.1.0 was used to quantify the protein bands.

### 2.8. Cell Culture

The immortalized human EVT cell line HTR-8/SVneo and the human choriocarcinoma cell line JEG-3 were provided by Zhou Chen from State Key Laboratory. DMEM/F-12 and RPMI (Gibco, Waltham, MA, USA) were used to culture HTR-8/SVneo and JEG-3, respectively, supplemented with 12% fetal bovine serum (FBS). Furthermore, the human umbilical vein endothelial cell line Huvec was obtained from Shanghai Cell Bank (Beyotime Biotechnology, Shanghai, China) and cultured in a DMEM medium. All cells were cultured in a humidified atmosphere with 5% CO_2_ at 37 °C.

### 2.9. 5-Ethynyl-2′-deoxyuridine (EdU) and Terminal Deoxynucleotidyl Transferase dUTP Nick end Labeling (TUNEL) Assay

To assess the replication of DNA and cell proliferation, the EdU assay kits (RiboBio, Guangzhou, China) were utilized. The cell under logarithmic growth was seeded into 48-well plates (0.4 × 10^4^) and cultivated with F-12/DMEM medium including 12% FBS for 36 h. Then, 4% formaldehyde-disposed cells gained fixation with further penetrations in 0.4% TritonX-200 for 15 min at 27 °C. Consequently, treated cells were dyed with the Hoechst and EdU based on instructions from the manufacturer. A TUNEL assay kit was utilized for assessing cell apoptosis. Referring to the instructions, the differentially processed cells were implanted in a 6-well plate with the same density. After 24 h, the plates experienced washing, fixation, permeation, and incubation. Ultimately, images were captured by a fluorescence microscope.

### 2.10. Flow Cytometry

About the samples applied to detect apoptosis, we dealt with the following processes. First, a 15 mL centrifugal tube containing the entire medium was collected and digested with 0.25% trypsin. The cells were treated with the subsequent well combination and centrifuged at 2000 revolutions per minute for five minutes. The supernatant was discarded and replaced using brand-new precooled PBS. These actions were carried out twice. The cell pellet was consequently broken in 1000 mL of PBS. The sole difference in terms of studying the cell cycle was that 500 mL of iced 75% ethanol was used to resuspend the cell pellet. A flow cytometer (CytoFLEX, eBioscience, San Diego, CA, USA) was employed following the manufacturer’s recommendations to evaluate the indications of cell apoptosis and cycles.

### 2.11. Wound-Healing Assay

To evaluate the migratory capabilities of human trophoblast cells, wound healing experiments were used. In a nutshell, JEG-3 cells that had been transfected were plated in 6-well dishes and grown till 95% confluent. Then, a 12 mL pipette tip was utilized to design a wound on the monolayer of the cell. A fresh medium including 2% fetal bovine serum was added to the wells after gently washing each well twice with PBS. A camera mounted on a light microscope was used to record the wound widths at 0 h, 12 h, 24 h, 36 h, and 48 h. To analyze the data, Image J was used (Version 1.46r).

### 2.12. Transwell Assay

A 1.5 mg/mL Matrigel was applied to the upper partition of the transwell with 10 m pores, and it was incubated for 5 h at 37 °C (BD Biosciences, Franklin Lakes, NJ, USA). After seeding transfected JEG-3 cells in 200 L of serum-free DMEM/F12 media, the lower section was loaded with 650 L of DMEM/F12 medium including 12% fresh FBS. After 36 h of incubation in 5% CO_2_ conditions at 37 °C, invasive cells were confirmed to have entered the matrix, whereas cells with the ability to migrate crossed over to the side of the membrane that was facing the lower chamber. Paraformaldehyde 4% was utilized to fix invading cells and 0.1% crystal violet (Beyotime Biotechnology, Shanghai, China) was utilized to stain them. The invasive cells were seen under a microscope (ZEISS, Aalen, Germany).

### 2.13. Plasmid Transfection

Either the control vector or the *PLK1* overexpression plasmid was transfected into the cells of HTR-8/SVeo. HTR-8/SVeo cells were planted in 12-well plates one day prior to transfection. One hour prior to obtaining 60–70% cell confluency, the preceding medium was changed to 900 mL of new media containing 12% FBS. Before being put into the 12-well plate, the previously described PolyJet and plasmids were blended with serum-free media (SignaGen, Frederick, MD, USA). The medium was switched out for a new medium after 10 h. For use in subsequent investigations, proteins or genes were extracted after two days.

### 2.14. Lentivirus Infection

Genechem provided lentiviruses containing human *PLK1* short hairpin RNA coding sequences (Shanghai, China). The short small interfering RNA (siRNA) targeting *PLK1* had the sequence 5′-GAACUAUGAGCAGAGAAUATT-3′. The infection index of the lentivirus was found to be 60 in early trials. According to the manufacturer’s recommendations, lentivirus transfection was performed on JEG-3 cells when they were 30–40% confluent. To find stable clones of transfected cells after two days, the transfected cells were cultured in 3 g/mL puromycin for a week. After that, cells were collected for Western blotting and other experiments.

### 2.15. Animal Preparation and RPL Model

The Ethics Committee and Animal Care Institution of King Faisal Specialist Medical University authorized all of the experimental protocols. All of the mice were maintained in a specified pathogen-free (SPF) room with a 12 h dark/light cycle, regular rodent food, and water access. A total of 27 7–9-week-old mice were superovulated by a pregnant mare of 12 IU gonadotropin intraperitoneally and then 12 IU hCG 40–50 h subsequently. After that, sex-ready male mice and female mice were kept in a cage at a balance of 1:2. Mice wearing vaginal pins were regarded as having given birth 0.5 days earlier. These mice were slaughtered and related tissues were removed at 8:00 each time 1.5 days after pregnancy. The samples were delivered right away to the lab. Using a stereo microscope to divide the fallopian tubes, two-cell phase embryos were harvested. In addition, these embryos were grown in a proportional M16 medium at 37 °C with 5% CO_2_ after being washed in phosphate-buffered saline with 1% bovine serum albumin (Sigma, St. Louis, MO, USA). At the two-cell phase, the embryos were processed using the *PLK1* inhibitor BI2536 (Beyotime Biotechnology, Shanghai, China). The pre-implantation developmental rate was calculated at the four-cell stage.

### 2.16. Immunofluorescence Staining of Embryos

After being fixed in 5% paraformaldehyde for 30 min, the processed embryos were permeabilized in phosphate-buffered saline including 0.6% Triton X-200 for 15 min at 25 °C. To prevent nonspecific binding, the embryos were put into microdroplets including 10% bovine serum albumin for two hours at 25 °C. After treatment with the primary antibody the next day, the embryos were flushed in 2% bovine serum albumin before being treated for two hours with the complementary secondary antibody. Then, the nuclei were dyed with Hoechst for 45 min. After that, confocal microscopy was used to take several pictures. The intensity of the fluorescence was measured with Image J.

### 2.17. Mitochondrial Superoxide Measuring

Utilizing MitoSox Red (Carlsbad, CA, USA), which was liquefied in a 1:1 combination of M16 and dimethylsulfoxide to a definitive concentration of 5 M, four-cell stage embryos were examined to identify the superoxide production in the mitochondria of the embryo. Embryos completed receiving the MitoSox intrathecal injection about 40 min later. Hoechst was then used to stain for 30 min in complete darkness. A confocal laser scan microscope was used to survey the red fluorescence at 510/580 nm. reactive oxygen species (ROS) levels were assessed by the intensity of fluorescence.

### 2.18. Potential of Mitochondrial Membrane Determination

A lipophilic cationic dye called JC-1 was able to enter the mitochondrial matrix while the membrane of the mitochondria became polarized. The dye may traverse the membrane and aggregate into J-aggregates, which may emerge red when exposed to Ultraviolet light if the mitochondrion has membrane potentials (m). When low m is present, the dye maintains its monomeric state, and the fluorescence shows green. Fluorescence microscopy makes it simple to examine the various distributions of red and green, and ratio labeling of red-to-green is utilized to evaluate the mitochondrial membrane potential in mouse embryos. The embryos were then incubated for 25 min with Hoechst (Beyotime Biotechnology, Shanghai, China), flushed with 1% BSA, and seen using a confocal microscope. To assess statistically sound data, the fluorescence intensities were documented, and three separate tests were carried out.

### 2.19. Statistical Analysis

Prism v.8.0 (GraphPad Software Inc, San Diego, CA, USA) and SPSS 23.0 were used for creating the graphs and for the analysis of data. Quantitative data are presented as the mean ± SEM, and the data were statistically evaluated using Student’s t-tests. A value of *p* < 0.05 was considered statistically significant.

## 3. Results

### 3.1. RPL Regulatory Network Construction and Hub High Traffic Genes Detection

A regulatory network containing 8829 edges and 1557 genes was created. Regulatory interactions between several genes are listed in Appendix A. With the help of MCODE’s comprehensive clustering, we detected a core module with 409 edges and 61 genes (Appendix A).

We identified 14 genes (*PLK1* as a hub high-traffic gene and *MMP2*, *MMP9*, *BAX*, *MFN1*, *MFN2*, *OPA1*, *COX15*, *FOXO1*, *BCL2*, *DRP1*, *FIS1*, *TRAF2*, and *TOP2A*) in RPL regulatory network.

### 3.2. Pathway Enrichment Analysis

Forty-three pathways were identified in the RPL regulatory network, according to the findings (Figure 1). The NF-κB signaling, FOXO signaling, PI3K/AKT pathway, and endometrial cancer pathway were shown to be the most significant pathways in the RPL regulatory network.

### 3.3. Biological Processes and Protein Expression

Western blot examination was operated to assess the protein expression of PLK1, OPA1, MFN1, and DRP1 from healthy individuals and patients (Figure 2A). Similar results were also seen in the qRT-PCR analysis’ output (Figure 2B). Immunohistochemistry examination showed that PLK1 was localized in the villi nucleoplasm and that PLK1 expression was significantly increased in the healthy group (*p* < 0.01) (Figure 2C).

### 3.4. PLK1 as Hub High Traffic Gene Exerts Positive Effect on the Cell Differentiation

Following the findings of qRT-PCR analysis (*p* < 0.01), WB analysis indicated that the PLK1 expression in the villi tissue of RPL patients dropped by 76% when compared to the normal group (Figure 2A,B). Collectively, these results indicated that *PLK1* is expressed in the villi and that *PLK1* expression is decreased in patients with RPL in comparison with that in the healthy group.

### 3.5. In Trophoblasts, PLK1 Knockdown Decreases Proliferation and Increases Apoptosis

We examined the expression of *PLK1* in four trophoblast cell lines including BeWo, JAR, HTR8/SVneo, and JEG-3. *PLK1* expression was high in JEG-3 cells and low in HTR8/Svneo cells. Hence, we selected these two cell lines as in vitro models to further explore the role of PLK1 in trophoblast functions. si-RNA targeting *PLK1* and a PLK1-overexpression plasmid were transfected into to JEG-3 and Htr8 cells, respectively, to generate PLK1 knockdown (si-PLK1) and PLK1-overexpression (OE-PLK1) cells. The efficiency of knockdown and overexpression was verified by qRT-PCR and Western blot (WB) (*p* < 0.05) (Figure 3A). WB analysis indicated that PLK1 expression in the si-PLK1 group decreased by 93% compared with that in the si-NC group (*p* < 0.01), which was consistent with the qRT-PCR analysis (FC = 0.631) (*p* < 0.05). Additionally, PLK1 protein levels in Htr8 cells in the OE-PLK1 group were 1.3 times higher than that in the vector group, which was consistent with the results of the qRT-PCR (FC = 1.31) (*p* < 0.05). In terms of proliferation, EdU, a thymidine analog that is incorporated into DNA during its synthesis, was used to assess cell proliferation. PLK1 knockdown considerably reduced the proportion of EdU-labelled cells, suggesting downregulation of cell proliferation. The expression of TRAF2 increased in si-PLK1 JEG-3 cells (*p* < 0.05), while it decreased in OE-PLK1 HTR8/Svneo cells (*p* < 0.01) (Figure 3B). Furthermore, the apoptosis ratio was also assessed with flow cytometry; overexpression of PLK1 reduced the apoptosis ratio compared to that in vector-transfected cells (*p* < 0.01), whereas the opposite was observed in the si-PLK1 group (*p* < 0.01) (Figure 3C,D). Fluorescence intensities in the TUNEL assay also supported these findings (Figure 3E).

### 3.6. PLK1 Promotes Trophoblasts’ G1-S Transition

We performed flow cytometry to analyze the effect of dysregulated PLK1 expression in the cell cycle (Figure 4A,B). Overexpression of PLK1 in Htr8 cells resulted in a significant increase in the percentage of cells in the S phase (*p* < 0.05), hereas downregulation of PLK1 in JEG-3 cells resulted in the G1 phase (*p* < 0.001). As for the G2 phase, no difference was observed in the PLK1 knockdown or plasmid-transfected groups.

### 3.7. Plk1 Stimulates Trophoblast Migration and Invasion

Further research was conducted to specify whether PLK1 can affect the invasion and migration of JEG-3 cells using wound healing and transwell assays. As seen in Figure 5A, there were significantly fewer invaded and migrating cells in the si-PLK1 group relative to the si-NC group (*p* < 0.01). Real-time PCR and Western blots were used to analyze PLK1’s impact on invasion potential. When compared to the si-NC group, the MMP-9 and MMP-2 expression, which stimulates invasion, was decreased in the si-PLK1 group (*p* < 0.001) (Figure 5B). PLK1 was found to have a favorable impact on invasion and migration as a result. Collectively, these findings suggest that a lack of PLK1 might cause early placental implantation, which would contribute to RPL to a certain extent.

### 3.8. PLK1 Stimulates the Trophoblast Cell Cycle and Proliferation

To further investigate whether PLK1 mediates its functions via the NF-κB pathway, the expression of NF-κB-pathway-related proteins and genes were measured. In comparison to the si-NC group, the si-PLK1 group had higher levels of NF-κB and TRAF2 after PLK1 knockdown (*p* < 0.05) (Figure 6A). Our findings demonstrated that PLK1 controls NF-κB signaling pathway activity to control proliferation and apoptosis. We utilized the Htr8 cells to check the downstream signaling species expression in order to further support these findings. PLK1-overexpression (OE-PLK1) significantly reduced the expression of TRAF2 (Figure 6B). In addition, the expression of cell-cycle-associated proteins including Foxo1, cyclinD3, and CDK6 was assessed using WB. The expression of FoxO1, CDK6, and cyclinD3 was significantly upregulated in the OE-PLK1 group compared to that in the si-PLK1 group (*p* < 0.05) (Figure 6C). Collectively, these results imply that the NF-κB pathway plays a regulatory function in PLK1’s regulation of cell apoptosis, proliferation, and cycle in JEG-3 and Htr8 cells.

### 3.9. Mice Blastocyst Development and Trophectoderm Differentiation Are Both Negatively Regulated by the PLK1 Inhibitor

Figure 7A shows the procedure followed to acquire and discard pre-implantation embryos. We cultivated two-cell embryos in a medium including BI2536, a specific inhibitor of PLK1, to better understand the function of PLK1 in the mice trophectoderm (TE) differentiation and blastocysts maturation. We verified that the BI2536-treated group’s blastocyst-developing and four-cell rates were significantly lower (Figure 7B). Based on detectable and exclusive staining of NANOG and GATA4, total cell counts from DAPI staining were categorizedas those corresponding to cells of the c inner cell mass or the trophectoderm (TE); TE, which are the outer cells that are not considered as part of the inner cell mass of the blastocyst, did not exhibit NANOG and GATA4 staining. The result showed that trophectoderm differentiation of the control group was higher than the BI2536-treated group (*p* < 0.01) (Figure 7C).

### 3.10. Expression of PLK1-Induced ROS and Dysfunction of the Mitochondria

We investigated whether PLK1 suppression led to ROS accumulation, which therefore slowed down the development of the embryo. Utilizing a confocal microscope, mitochondrial superoxide—a measure of the amount of ROS—was explored in embryos to investigate the inhibition mechanism of BI2536-treated PLK1. As seen in Figure 7D, four-cell embryos treated with BI2536 had significantly higher amounts of mitochondrial superoxide than the control embryos (*p* < 0.05). We stained JC-1 to analogize the fluorescence intensity in the BI2536-mediated and unmediated groups since ROS buildup can impair mitochondrial membrane permeability. When compared to the control group, embryos treated with BI2536 displayed a significantly higher level of green fluorescence, which indicated a reduction in the potential of the mitochondrial membrane (*p* < 0.05) (Figure 7E). The proteins expression of mitochondrial fission including FIS1 and DRP1 was decreased (Figure 2B). These findings imply that PLK1 reduced NF-κB activity and reduced apoptosis, high permeability, and mitochondrial dysfunction in embryo cells.

## 4. Discussion

Regarding the critical role of nodes with a high degree (hubs) in preserving the integrity of the network construction in anticipation of attacks and failures [30,31] in disseminating events [32], it is realistic to anticipate that hub nodes/genes control is crucial to prevent disease networks. Some hub genes cause lethality or infertility, making them unsuitable drug targets. Placental trophoblast cells can promote embryo implantation, uterine spiral artery remodeling, and placentation via their proliferation, invasion, and migration. Furthermore, these cells secrete numerous active substances to regulate maternal–fetal interactions to ensure the normal growth and development of the embryo and fetus [33]. It is important to understand the changes in trophoblast cell function in RPL to explain its pathogenesis. Whole-exome sequencing of villi indicated that different pathogenic genes played a vital role in RPL. For example, one patient was found to have compound heterozygous mutations in or jointly involved in the occurrence of RPL by affecting inflammation, oxidative stress response, and angiogenesis [34,35] at the maternal–fetal interface. Transcriptomics is widely used in the study of villi tissue of RPL. The RNA sequencing results showed that different genes’, for example, *EGR1*, *PDLIM1*, and *MAPK3*, mRNA expression levels were reduced in the trophoblasts of RPL [36]. Another study found that the most differentially expressed genes (DEGs) in trophoblastic cells of RPL could bind the transcription factor E2F [37]. Some studies have focused on the role of ncRNAs in RPL, and the differentially expressed ncRNAs participate in biological pathways, including immunity, apoptosis, and hormonal regulation [38,39]. Transcriptome studies have revealed that the altered expression of genes or regulatory factors was associated with trophoblast proliferation, invasion, migration, and apoptosis, resulting in placental dysfunction and embryo failure to survive. In addition, the function of trophoblasts during pregnancy is affected by the immune response at the maternal–fetal interface. DEGs involved in the balance between pro- and anti-inflammatory responses also contribute to the development of RPL.

Omics of the villi of RPL patients can indicate new biomarkers for the diagnosis and treatment of RPL; therefore, in this study, we constructed an RPL gene network using omics levels. In this regard, 14 genes in the RPL regulatory network were predicted to be involved in RPL-related biomarkers and risk prognosis. We studied whether PLK1 is engaged in RPL pathogenesis by controlling the embryo’s development and trophoblast cell functions and PLK1’s effect on the alteration of the chorionic villi transcriptome. We identified *PLK1* as a hub high-traffic gene involved in RPL. We revealed that expression of *PLK1* is decreased in the villi of RPL patients. *PLK1* was previously found to be significantly expressed in hepatocellular carcinoma and to have a favorable impact on both cancer cell invasion and proliferation, according to Ando et al. [40]. Additionally, exome sequencing in unrelated samples or single cases has also identified candidate maternal-effect genes, including *PIF1*, *CCDC68*, *PLK1*, *MMP10*, *FLT1*, *PADI6*, and *FKBP4* [41,42,43,44,45]. Forkhead box transcription factors are well known to have a function in the regulation of a wide range of biological procedures, such as oxidative stress, DNA damage and repair, cell proliferation, apoptosis, and cell differentiation, all of which are extremely important for cell biology [46].

In this study, we also demonstrated that PLK1 guards against RPL by balancing mitochondrial fission and fusion, lowering the apoptosis of chorionic villi. Additionally, our results showed that PLK1’s protective impact depends on its negative control of NF-κB signaling. The investigation also found that *PLK1* limits the *TRAF2* expression to control NF-κB signaling by interacting with *TRAF2*, an upstream NF-κB regulator. The PLK1/ TRAF2/ NF-κB axis was therefore demonstrated to be essential in RPL and may represent a possible therapeutic target in the clinic. Mitochondria are crucial organelles in chorionic villi because their dysfunction can lead to loss of function in different organs, such as the heart, kidneys, and lungs [47,48]. The present study revealed that RPL-induced mitochondrial damage was characterized by increased ROS, decreased MMP, and disruption of mitochondrial dynamic balance in the chorionic villi. Mitochondrial dynamic balance depends on fission and fusion, which are mediated by conserved dynamin-related GTPase proteins including the fusion proteins optic atrophy 1 (OPA1), mitofusin1 (Mfn1), and mitofusin2 (Mfn2), as well as the fission protein dynamin-related protein 1 (Drp1) and its receptor mitochondrial fission protein 1 (Fis1) [49,50]. Our results showed that the expression of Fis1 and Drp1 was increased in the villi tissue during RPL, while Mfn2, Mfn1, and OPA1 expression was decreased, indicating increased mitochondrial fission and insufficient mitochondrial fusion. Furthermore, impaired mitochondria release cytochrome c, an essential component of the respiratory chain, into the cytosol to trigger apoptosis [51,52]. The B-cell lymphoma 2 (Bcl2) family mediates cytochrome c release, and cytochrome c in the cytosol can bind to apoptotic protease factor 1 (Apaf1), forming the apoptosome complex, which activates caspase 9 and caspase 3, and resulting in apoptotic features such as DNA fragmentation and chromatin condensation [53]. This study also verified excessive apoptosis in the chorionic villi, which was characterized by elevated expression of Bax and cleaved caspase 3, decreased Bcl2 expression, and an increase in TUNEL-positive cells, which was accompanied by mitochondrial dysfunction during RPL. NF-κB signaling is known to play a pivotal role in inflammation by mediating the expression of inflammatory cytokines and chemokines [54]. The NF-κB family is composed of five members known as RelA/p65, RelB, c-Rel, NF-κB1 (p105/p50), and NF-κB2 (p100/p52), among which p50/p65 is the most representative dimer [55]. Incremental mitochondrial fragmentation can in turn increase NF-κB activity by phosphorylating IKK and IκBα through ROS accumulation [56]. Similarly, the present results demonstrated that pharmacological inhibition of the NF-κB pathway not only ameliorated the changes in fission proteins (Drp1 and Fis1) and fusion proteins (OPA1, Mfn1, and Mfn2) but also improved mitochondrial function by rescuing MMP in the chorionic villi during RPL.

Moreover, a TRAF-binding protein called TANK was discovered to have opposing regulatory features in innate immune activation. Because TANK can increase the NF-κB signaling function that expresses TRAF2, the impact of the inhibition of NF-κB and TRAF2 on the NF-κB pathway is also debatable. We showed at the TRAF2 level in the RPL and confirmed that the expression of TRAF2 was noticeably enhanced both in vivo and in vitro during RPL since the expression of TRAF2 varied with various stimuli [57].

We utilized BI2536 (S1109), a PLK1 inhibitor, in this investigation to further confirm the impact of PLK1 on the quality and growth of the pre-implantation embryo and trophoblast functions. Two cell embryos were cultured in an M16 medium containing BI2536 to observe the effect of PLK1 inhibition on four-cell embryos and blastocysts. We observed that embryos treated with BI2536 showed a reduced developmental rate compared to the control group, indicating that PLK1 inhibition negatively affects embryo quality. Furthermore, pooled embryos showed decreased differentiation of TE in comparison to the healthy group, suggesting that PLK1 may hinder the growth and functions of TE, thereby contributing to RPL. In oocytes and early embryos, mitochondria play a pivotal role in ATP generation as mitochondrial-based oxidative metabolism is required for development, rather than glycolysis [58]. The main function of mitochondria is ATP generation, which is directly associated with the potential of embryo development [59]. Apart from increased rates of aneuploidy, mitochondrial dysfunctions, including oxidative damage, changes in mitochondrial membrane potential, and decreased ATP generation, are observed in oocytes and embryos of older mothers, compared with those of their younger counterparts in both animal and human models [60,61,62,63], indicating that reduced rates of blastocyst development can be partly attributed to metabolic dysfunction. Following further investigation, it was discovered that PLK1 and TRAF2 physically interact in RPL, providing structural proof that PLK1 regulates the activity of NF-κB signaling by TRAF2. We observed that inhibition of PLK1 can suppress trophoblast proliferation, invasion, and migration through the NF-κB signaling pathway and impair the development potential of pre-implantation embryos by dysregulating mitochondrial functions. Therefore, we proposed that PLK1 binds to TRAF2 during RPL and inhibits it, negatively regulating NF-B signaling.

The study limitations should be mentioned. The experimental period was relatively short, and the collected samples were limited by region.

## 5. Conclusions

We proposed the construction of a recurrent pregnancy loss regulatory network and identified four related pathways of RPL. As well, 14 genes and proteins were detected, which can act as therapeutic targets for recurrent pregnancy loss treatment and further investigations. These results also revealed a detailed mechanism of *PLK1* as a hub high-traffic gene in regulating the proliferation, migration, and invasion of cells in RPL. Our research verified that PLK1 prevents RPL by regulating mitochondrial malfunction and preventing apoptosis via the TRAF2/NF-κB signaling pathway. Therefore, the PLK1/TRAF2/NF-κB axis can be used as a potential clinical target for the prevention and treatment of RPL.

## Figures and Tables

**Figure 1 biomedicines-11-00879-f001:**
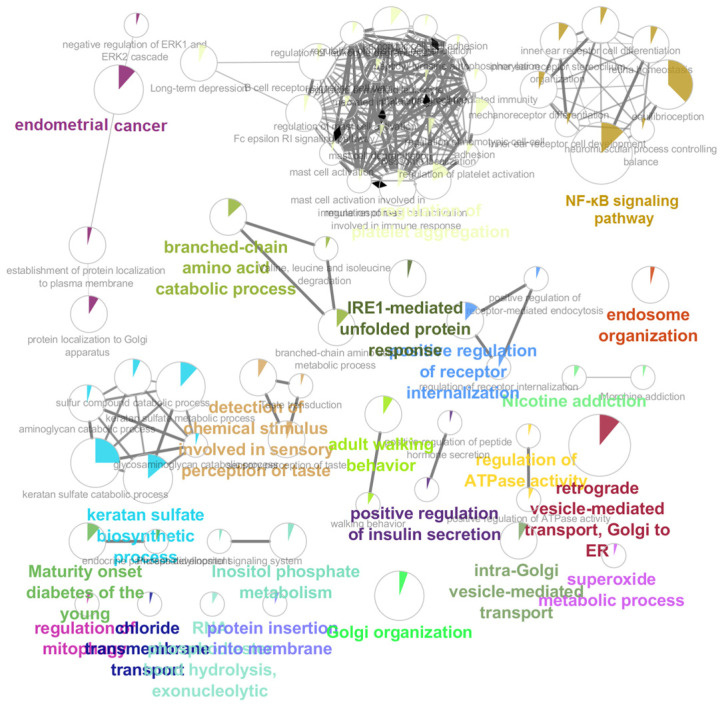
GO and KEGG pathway analyses between the villi samples in RPL patients and the healthy group. The size of the nodes was highly correlated with the genes’ number, and nodes gradually increased from small to large. Different colors represent various biological processes.

**Figure 2 biomedicines-11-00879-f002:**
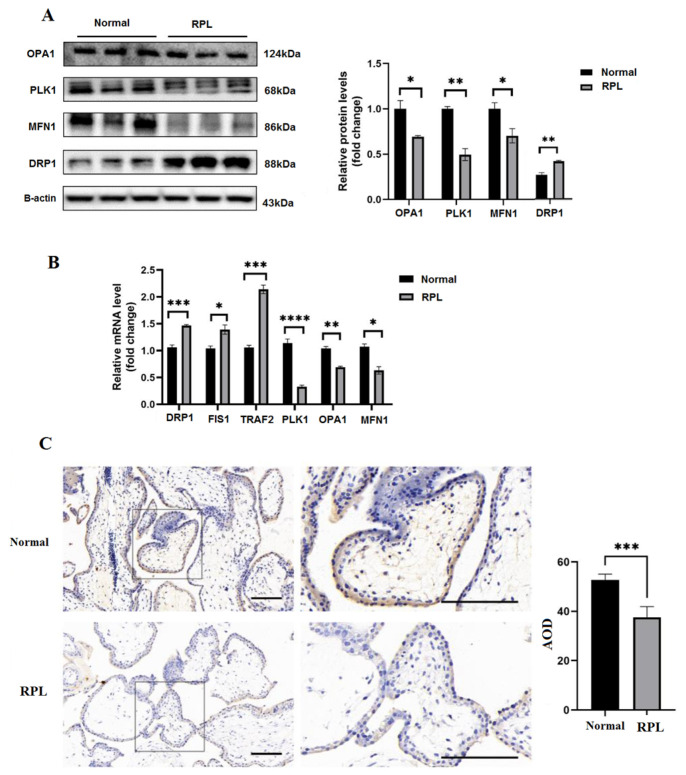
Hub genes dysregulation: (**A**) WB exhibited a change in the expressions of PLK1, DRP1, MFN1, and OPA1; (**B**) qRT-PCR analysis of hub genes, such as *PLK1*, *DRP1*, *TRAF2*, *MFN1*, and *OPA1*; (**C**) IHC staining images of the RPL samples and the healthy group; * *p* < 0.05, ** *p* < 0.01, *** *p* < 0.001, **** *p* < 0.0001.

**Figure 3 biomedicines-11-00879-f003:**
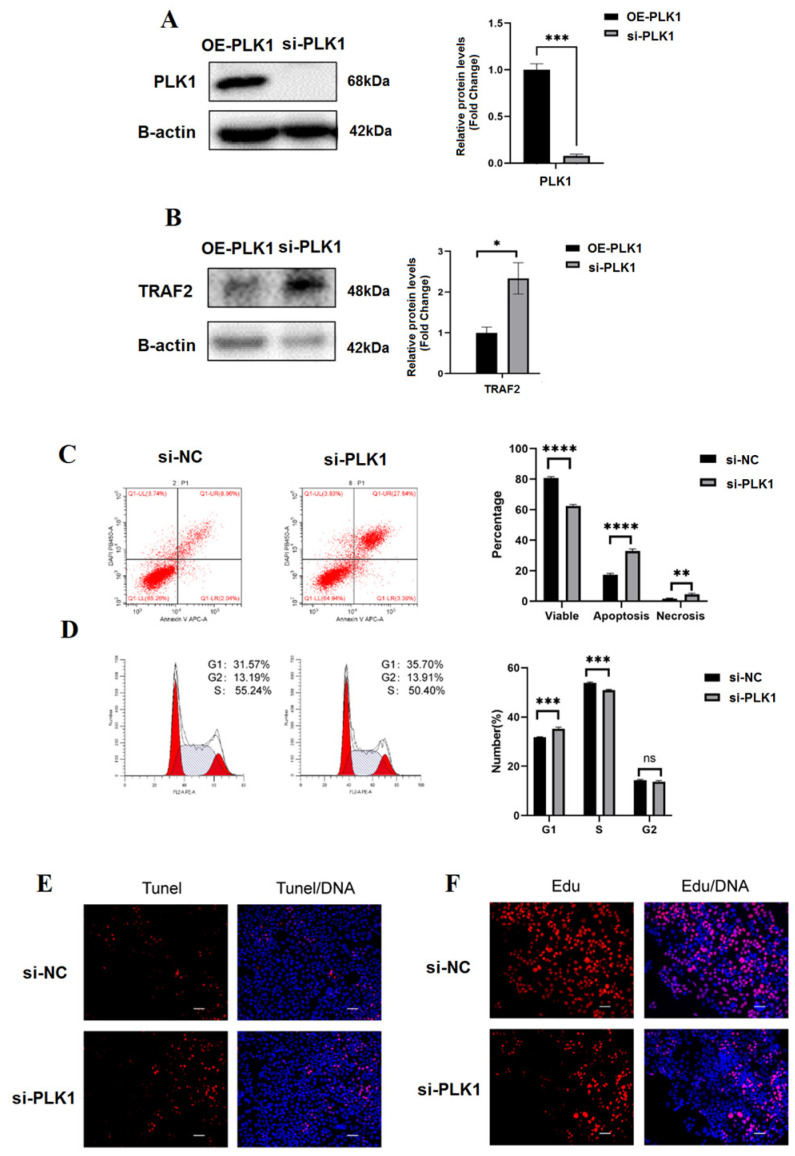
PLK1 knockdown arrests the cell cycle and suppresses proliferation and trophoblasts apoptosis: (**A**) PLK1 expression was examined following transfection of particular si-RNA and OE-PLK1; (**B**) WB analysis was utilized to detect how PLK1 knockdown affected the amount of protein in the TRAF2 protein; (**C**) flow cytometry was utilized to measure apoptosis; (**D**) cell cycle modifications in JEG-3 cells after treatment; (**E**) yo evaluate the impact of PLK1 knockdown on apoptosis, the TUNEL assay was utilized; and (**F**) using the EdU assay, the PLK1 effects on cell proliferation were evaluated; * *p* < 0.05, ** *p* < 0.01, *** *p* < 0.001, **** *p* < 0.0001.

**Figure 4 biomedicines-11-00879-f004:**
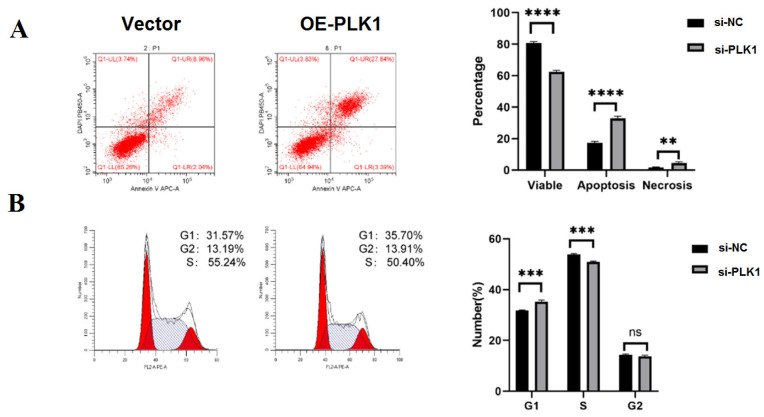
Overexpression of PLK1 suppressed apoptosis and stimulates the proliferation of cells in TE; (**A**) flow cytometry analysis was utilized to measure apoptosis; and (**B**) a flow cytometry study was conducted to compare the two groups’ differing cell cycle patterns; ** *p* < 0.01, *** *p* < 0.001, **** *p* < 0.0001.

**Figure 5 biomedicines-11-00879-f005:**
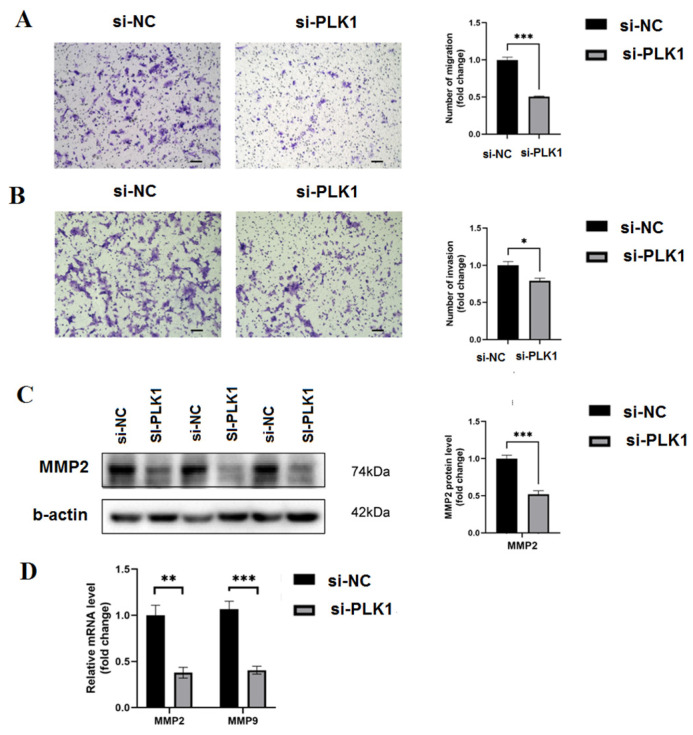
PLK1 stimulates trophoblast cells to invade and migrate: (**A**) the transwell assay was performed to measure the trophoblasts’ migration; (**B**) the transwell test was used to measure trophoblast invasion; (**C**) the impact of PLK1 on MMP-2 protein expression; and (**D**) qRT-PCR assesses how *PLK1* knockdown affected the levels of *MMP2* and *MMP9* mRNA; * *p* < 0.05, ** *p* < 0.01, *** *p* < 0.001.

**Figure 6 biomedicines-11-00879-f006:**
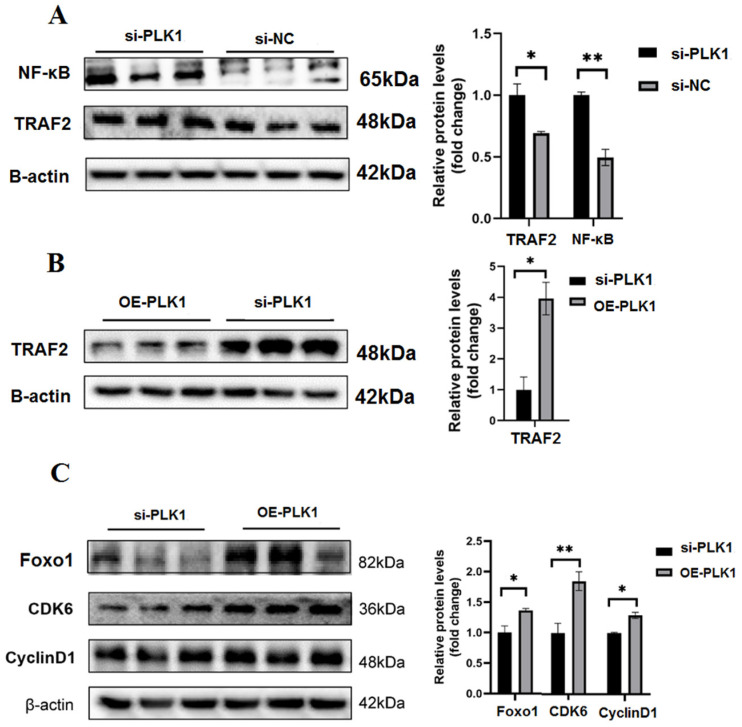
(**A**) The NF-κB signaling pathway is impacted by PLK1 inhibition. After PLK1 downregulation, NF-κB and TRAF2 levels changed, as seen in the Western blotting analysis; (**B**) WB to evaluate alterations in the levels of TRAF2 in the si-PLK1 and OE-PLK1 group; (**C**) WB to evaluate alterations in the levels of FoxO1, CDK6, and cyclinD3 in the si-PLK1 and OE-PLK1 group; * *p* < 0.05, ** *p* < 0.01.

**Figure 7 biomedicines-11-00879-f007:**
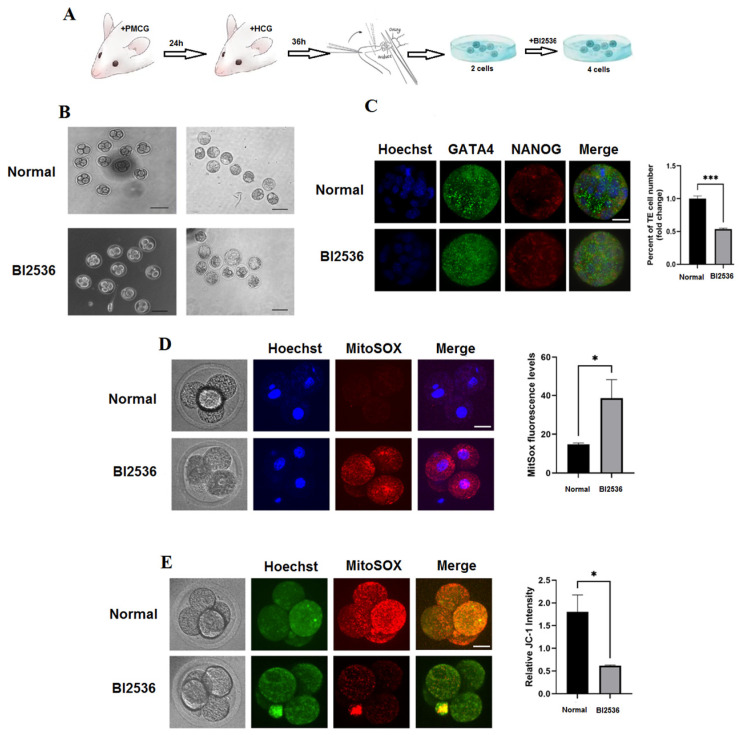
PLK1 inhibition affects the quality and potential for the development of pre-implantation embryos: (**A**) flowchart illustrating how embryos are collected; (**B**) the PLK1 inhibitor limited the growth of blastocysts and four-cell embryos; (**C**) total ICM staining, Hoechst nucleus/DNA staining, NANOG staining, and GATA4 primitive endoderm staining (GATA4 and NANOG merged); (**D**) how BI2536 affects the mitochondrial superoxide concentrations in mice four-cell embryos. To determine how BI2536 impacts mitochondrial potential, € mice four-cell embryos were stained with JC-1. Scale bar of 20 m. * *p* < 0.05, *** *p* < 0.001.

## Data Availability

The National Center for Biotechnology Information’s BioProject and GEO datasets were utilized. GSE180485, GSE161969, and GSE65102 are the corresponding dataset accession numbers.

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
