# Peer review of "Designing Effective Multi-Target Drugs and Identifying Biomarkers in Recurrent Pregnancy Loss (RPL) Using In Vivo, In Vitro, and In Silico Approaches"

_biomedicines, 2023, doi:10.3390/biomedicines11030879_

Round 1

Reviewer 1 Report

The manuscript " Design Effective Multi-target Drugs and Identify Biomarkers 2 in Recurrent Pregnancy Loss (RPL) Using In Vivo, In Vitro, and 3 In Silico Approaches " is an interesting work which aims to identify hub genes and proteins involved in the recurrent pregnancy loss (RPL) biological process. The work is original and well-structured. The design of the project is appropriate and the results are significant. The statistical analysis is well conducted and the language is acceptable. It is suitable for publication in the present form

Author Response

The manuscript " Design Effective Multi-target Drugs and Identify Biomarkers 2 in Recurrent Pregnancy Loss (RPL) Using In Vivo, In Vitro, and 3 In Silico Approaches " is an interesting work which aims to identify hub genes and proteins involved in the recurrent pregnancy loss (RPL) biological process. The work is original and well-structured. The design of the project is appropriate and the results are significant. The statistical analysis is well conducted and the language is acceptable. It is suitable for publication in the present form.

Dear Reviewer:

Thank you for the comments concerning our manuscript entitled ‘Design Effective Multi-target Drugs and Identify Biomarkers in Recurrent Pregnancy Loss (RPL) Using In Vivo, In Vitro, and In Silico Approaches’.

Reviewer 2 Report

In the article entitled: Design Effective Multi-target Drugs and Identify Biomarkers in Recurrent Pregnancy Loss (RPL) Using In Vivo, In Vitro, and In Silico Approaches authors have tried to investigate the important problem from the highly developed society the pregnancy pathology. Their efforts were concentrated around polo-like kinase 1 and the axis between them and TRAF2-NFkB. The idea was an interesting and valuable investigation due to the assistance of mitochondrial dysfunction and apoptotic processes in the light of RPL. From the technical point of view, all presented techniques are correctly selected. However, the number of patients is too small to have some statistical importance for 26 women aged between 19-40 years old. The group size is too small to get valuable results, it is like reading tea leaves. Moreover, in the whole article, I did not find any information about ethical and bioethical statement numbers - it is obligatory for the experiment with human tissue taken from patients and for experiments with animals. Additionally,y I strongly recommend rewriting the article due to the sentence overlength which makes the idea difficult to follow. In the title, the phrase preliminary studies should be introduced.

In conclusion, I can not recommend this article for publication before correction.

Author Response

Dear Reviewer:

Thanks to the reviewer’s specialistic comments, those comments are all valuable and very helpful for revising and improving the quality of our manuscript, as well as the important guiding significance to our research. According to your advice, we have amended the relevant part of the manuscript. We evaluated approximately 143 samples in different databases before starting the main analysis because sample quality and number are the most important parameters that affect the accuracy of results. These methods used in this study depend on sample quality. As well, we had been deeply aware of the limitations of our sample size and emphasized the deficiencies in the discussion section. Therefore, we have used more data sets for increasing the accuracy of results.

Point 1: Moreover, in the whole article, I did not find any information about ethical and bioethical statement numbers - it is obligatory for the experiment with human tissue taken from patients and for experiments with animals.

Response 1: We have added ethical and bioethical statement numbers to the material and method section.

Point 2: Additionally, I strongly recommend rewriting the article due to the sentence overlength which makes the idea difficult to follow. In the title, the phrase preliminary studies should be introduced.

Response 2: We sincerely apologize for the inaccurate grammar of the manuscript. We would like to express our sincere gratitude for the reviewer’s rigorous criticism. The manuscript has been revised and proofread by two English native speakers involved in medical research. As the reviewer kindly mentioned, We have further defined and introduced the phrase preliminary.

Point 3: In conclusion, I can not recommend this article for publication before correction.

Response 3:  We have tried to revise the manuscript according to your advice, and we have amended the relevant part of the whole manuscript.

Reviewer 3 Report

In this manuscript the authors want to understand the molecular mechanism responsible for the recurrency pregnancy loss (RPL). 

I understand that is difficult to summarize in a reduced number of characters all the performed experiments, but the abstract is not simple to understand and should be rewritten in a clearer way. 

In Introduction, lanes 72-74, it’s not clear what the authors want to say…is NF-kB pathway crucial or its role is unclear and debatable?

In Results, the authors should write a conclusive sentence describing the obtained results and their correlation with the aim of this research work.

In Par 3.5 and image 2 A and B, the authors observe and comment that PLK1 expression is higher in healthy group, but they don’t comment why the expression of the other proteins is different between healthy and RPL groups (OPA1 and MFN are higher in healthy group while DRP1 is lower and the difference in expression of MFN and DRP1 between the two groups is considerable). The authors should discuss and comment these results and describe the role of these proteins thus justifying the choice. 

The authors use different cell lines but they don’t specify the origin of these.

Lanes 329-331 the authors should better describe their transfection experiments. Did you perform 3 or 2 different transfections?

Lanes 331-333. The authors assert that in Fig 3A we can observe the transfection efficiency both for siRNA and for PLK1 OE but the fig only represents the PLK1 level in siRNA experiment.

Please specify in text, fig legends and materials and methods which cell line is used in each experiment.

Except for the analysis of the apoptosis, lacks all the images linked to the PLK1 overexpression. 

Did the authors analyze the cell cycle in cells with the PLK1 overexpression? All the effects linked to the overexpression should be shown.

Lane 407, the authors mention results obtained with the use of PDTC but they don’t show anything even if mention one fig 2B that is not clear which is.

Furthermore, the authors assert that PLK1 works by altering the NF-kB translocation, but this is never demonstrated.

In sum, the manuscript could be interesting and possess good data but a lot of these are not present. The authors should be modify all the manuscript paying attention to well describe all in detail and to not forget images and performing other experiments supporting the aim of their work.

Author Response

Dear Reviewer:

Thank you for the comments concerning our manuscript entitled ‘Design Effective Multi-target Drugs and Identify Biomarkers in Recurrent Pregnancy Loss (RPL) Using In Vivo, In Vitro, and In Silico Approaches’. Those comments are all valuable and very helpful for revising and improving the quality of our manuscript, as well as the important guiding significance to our research. According to your advice, we have amended the relevant part of the manuscript. The main corrections in the paper and the responses to the reviewer’s comments are as flowing:

Point 1: I understand that is difficult to summarize in a reduced number of characters all the performed experiments, but the abstract is not simple to understand and should be rewritten in a clearer way.

Response 1: We thank the reviewer for the insightful comments and valuable suggestions. We have tried to rewrite the abstract to simplify understanding.

Point 2: In Introduction, lanes 72-74, it’s not clear what the authors want to say…is NF-kB pathway crucial or its role is unclear and debatable?

Response 2: We are grateful for the reviewer’s constructive comments. We have rewritten the sentence, our means is “different biological processes and pathways.

Point 3: In Results, the authors should write a conclusive sentence describing the obtained results and their correlation with the aim of this research work.

Response 3:  We have tried to rewrite and add a conclusive sentence describing the obtained results and their correlation with the aim of this research work to the results section. We have also tried to interpret the findings comprehensively in the discussion section.

Point 4: In Par 3.5 and image 2 A and B, the authors observe and comment that PLK1 expression is higher in healthy group, but they don’t comment why the expression of the other proteins is different between healthy and RPL groups (OPA1 and MFN are higher in healthy group while DRP1 is lower and the difference in expression of MFN and DRP1 between the two groups is considerable). The authors should discuss and comment these results and describe the role of these proteins thus justifying the choice.

Response 4: We have discussed and added more relevant results. We have also tried to interpret the expression of the other proteins as different between healthy and RPL groups and their correlation with aim of the study. Therefore discussion section was improved completely. In this regard, some contexts were removed and some paragraphs were relocated or rewritten.

Point 5: The authors use different cell lines but they don’t specify the origin of these.

Response 5: We have added the origin of cell lines.

Point 6: Lanes 329-331 the authors should better describe their transfection experiments. Did you perform 3 or 2 different transfections?

Response 6: We are grateful for the reviewer’s constructive comments. We have tried to explain and clarify this sub-section.

Point 7: Lanes 331-333. The authors assert that in Fig 3A we can observe the transfection efficiency both for siRNA and for PLK1 OE but the fig only represents the PLK1 level in siRNA experiment.

Response 7: We would like to express our sincere appreciation to reviewers for their professional opinions. As the reviewer kindly mentioned, we have adjusted the relevant pictures and mistyping which occurred. We have replaced si-NC with OE-PLK1 in figure 3.

Point 8: Please specify in text, fig legends and materials and methods which cell line is used in each experiment.

Response 8: We have specified the origin of cell lines in the whole manuscript.

Point 9: Except for the analysis of the apoptosis, lacks all the images linked to the PLK1 overexpression.

Response 9: We have illustrated PLK1 overexpression in figure 2 and figure 6.

Point 10: Did the authors analyze the cell cycle in cells with the PLK1 overexpression? All the effects linked to the overexpression should be shown.

Response 10: We have added related cell cycle proteins figure as Figure 6C.

Point 11: Lane 407, the authors mention results obtained with the use of PDTC but they don’t show anything even if mention one fig 2B that is not clear which is.

Response 11: We used BI2536 (S1109), a PLK1 inhibitor, in this investigation to further confirm the impact of PLK1 on the quality and growth of the pre-implantation embryo and trophoblast functions. Therefore, we have revised and removed PDTC. The effect of BI2536, we have tried to explain and depict in figure 7.

Point 12: Furthermore, the authors assert that PLK1 works by altering the NF-kB translocation, but this is never demonstrated.

Response 12: We have revised and focused on the main objective, therefore rewritten the whole manuscript.

Point 13: In sum, the manuscript could be interesting and possess good data but a lot of these are not present. The authors should be modify all the manuscript paying attention to well describe all in detail and to not forget images and performing other experiments supporting the aim of their work.

Response 13: Thank you for your comments. As the reviewer kindly mentioned, we have added forgotten images and performing other experiments. And we have also removed extra explanations to simplify and increase the reproducibility of the manuscript.

Reviewer 4 Report

1) Abstract. The NF-κB signaling, NER, PI3K/AKT, and endometrial cancer signaling pathways were identified via the RPL regulatory network. PLK1 as hub high traffic gene and MMP2, MMP9, BAX, MFN1, MFN2, OPA1, COX15, BCL2, DRP1, FIS1, TRAF2, and TOP2A. Clinical samples were examined, and the results demonstrated that RPL patients had tissues with decreased PLK1 expression than women with normal ovaries (p < 0.01). In vitro, PLK1 knockdown boosted NF-κB signaling pathway and apoptosis activation while decreasing cancer cell invasion, migration, and proliferation (p < 0.05). Furthermore, in vivo tests proved that cell mitochondrial function and trophectoderm development are both hampered by PLK1 suppression. Please, underline the most important statistically significant values to support the results.

2) Abstract. Our research shows that the PLK1/ TRAF2/ NF-κB axis is essential for RPL dysfunction and could be used as a clinical target for therapy. Please, improve the conclusions and underline the possible clinical implication of the study.

3) 1. Introduction. Please, better distribute the references to ameliorate the support of the different sentences.

4) 2.9. Cell Culture. Please improve this paragraph it is really important.

5) 2.20. Statistical Analysis. L 285-88. The Mean and SEM of the data, which were statistically examined applying the pro-grams GraphPad Prism 8 and SPSS 23.0, were provided. Each assay was carried out three times. The student t-test was utilized to assess the comparisons between the groups. Statistics were considered significant for p-values under 0.05. Improve this paragraph, underline how the values are reported and describe when different the statistical tests have been used.

6) Figure 1. GO and KEGG pathway analyses between the villi samples in RPL patients and healthy group. The size of nodes was highly correlated with the genes number and nodes gradually in-creased from small to large. Please improve the quality of the figure.

7) 3. Results. Underline in the text the most important statistically significant values to support the results.

8) 4. Discussion. Underline the limits of the study and create a separate paragraph.

9) 5. Conclusion. Improve this paragraph summarasing the most important results and pointing out possible clinical implication of the study.

Author Response

Dear Reviewer:

Thank you for the comments concerning our manuscript entitled ‘Design Effective Multi-target Drugs and Identify Biomarkers in Recurrent Pregnancy Loss (RPL) Using In Vivo, In Vitro, and In Silico Approaches’. Those comments are all valuable and very helpful for revising and improving the quality of our manuscript, as well as the important guiding significance to our research. According to your advice, we have amended the relevant part of the manuscript. The main corrections in the paper and the responses to the reviewer’s comments are as flowing:

Point 1: Abstract. The NF-κB signaling, NER, PI3K/AKT, and endometrial cancer signaling pathways were identified via the RPL regulatory network. PLK1 as hub high traffic gene and MMP2, MMP9, BAX, MFN1, MFN2, OPA1, COX15, BCL2, DRP1, FIS1, TRAF2, and TOP2A. Clinical samples were examined, and the results demonstrated that RPL patients had tissues with decreased PLK1 expression than women with normal ovaries (p < 0.01). In vitro, PLK1 knockdown boosted NF-κB signaling pathway and apoptosis activation while decreasing cancer cell invasion, migration, and proliferation (p < 0.05). Furthermore, in vivo tests proved that cell mitochondrial function and trophectoderm development are both hampered by PLK1 suppression. Please, underline the most important statistically significant values to support the results.

Response 1: We would like to express our heartfelt appreciation to the reviewer for the professional opinions. As the reviewer kindly mentioned, we have underlined the most important statistically significant values to support the results.

Point 2: Abstract. Our research shows that the PLK1/ TRAF2/ NF-κB axis is essential for RPL dysfunction and could be used as a clinical target for therapy. Please, improve the conclusions and underline the possible clinical implication of the study.

Response 2: We have tried to improve the whole abstract and the manuscript.

Point 3: Introduction. Please, better distribute the references to ameliorate the support of the different sentences.

Response 3: We have removed non-relevant references and have tried to add new relevant references and interpret them in the Introduction and the discussion section. In this regard, some contexts were removed and some paragraphs were relocated or rewritten.

Point 4: 2.9. Cell Culture. Please improve this paragraph it is really important.

Response 4: We have tried to improve this sub-section.

Point 5: 2.20. Statistical Analysis. L 285-88. The Mean and SEM of the data, which were statistically examined by applying the programs GraphPad Prism 8 and SPSS 23.0, were provided. Each assay was carried out three times. The student t-test was utilized to assess the comparisons between the groups. Statistics were considered significant for p-values under 0.05. Improve this paragraph, underline how the values are reported, and describe when different statistical tests have been used.

Response 5: We have tried to rewrite this sub-section.

Point 6: Figure 1. GO and KEGG pathway analyses between the villi samples in RPL patients and the healthy group. The size of nodes was highly correlated with the number of genes number and nodes gradually increased from small to large. Please improve the quality of the figure.

Response 6: We have increased the resolution of the Figure.

Point 7: 3. Results. Underline in the text the most important statistically significant values to support the results.

Response 7: We have emphasized the most important statistically significant values to support the results.

Point 8: Discussion. Underline the limits of the study and create a separate paragraph.

Response 8: Thanks to the reviewer’s specialistic comments, we have been deeply aware of the limitations of our sample size, and have emphasized the deficiencies in the discussion section.

Point 9: Conclusion. Improve this paragraph by summarizing the most important results and pointing out the possible clinical implication of the study.

Response 9: We thank the reviewer for the insightful comments and valuable suggestions. We have improved the conclusion based on the most important results.

Round 2

Reviewer 2 Report

The article can be accepted for publication. The corrections have been done accordingly to my remarks.

Reviewer 4 Report

No further comments